# Deletion of KCNQ2/3 potassium channels from PV+ interneurons leads to homeostatic potentiation of excitatory transmission

Heun Soh[1†], Suhyeorn Park[2†], Kali Ryan[1], Kristen Springer[1], Atul Maheshwari[2*], Anastasios V Tzingounis[1*]

[1]Department of Physiology and Neurobiology, University of Connecticut, Connecticut, United States; [2]Department of Neurology, Baylor College of Medicine, Texas, United States

*For correspondence:
atul.maheshwari@bcm.edu (AM);
anastasios.tzingounis@uconn.edu
(AVT)

[†]These authors contributed
equally to this work

Competing interests: The
authors declare that no
competing interests exist.

Reviewing editor: John
Huguenard, Stanford University
School of Medicine, United
States

**Abstract** KCNQ2/3 channels, ubiquitously expressed neuronal potassium channels, have emerged as indispensable regulators of brain network activity. Despite their critical role in brain homeostasis, the mechanisms by which KCNQ2/3 dysfunction lead to hypersychrony are not fully known. Here, we show that deletion of KCNQ2/3 channels changed PV$^+$ interneurons', but not SST$^+$ interneurons', firing properties. We also find that deletion of either KCNQ2/3 or KCNQ2 channels from PV$^+$ interneurons led to elevated homeostatic potentiation of fast excitatory transmission in pyramidal neurons. *Pvalb-Kcnq2* null-mice showed increased seizure susceptibility, suggesting that decreases in interneuron KCNQ2/3 activity remodels excitatory networks, providing a new function for these channels.
DOI: https://doi.org/10.7554/eLife.38617.001

## Introduction

Genetic studies have established that potassium channel dysfunction is responsible for multiple pediatric epilepsy disorders (*Brenner and Wilcox, 2012*; *Geisheker et al., 2017*; *Niday and Tzingounis, 2018*; *Oyrer et al., 2018*). KCNQ2 and KCNQ3 channels, in particular, have emerged as fundamental regulators of normal brain activity (*Brenner and Wilcox, 2012*; *Geisheker et al., 2017*). Pathogenic variants of these channels are strongly associated with early-onset neonatal epileptic encephalopathy and developmental disorders (*Millichap et al., 2017*; *Mulkey et al., 2017*; *Oyrer et al., 2018*). However, the network mechanisms by which *KCNQ2/3* variants lead to hyperexcitability are not fully understood.

KCNQ2/3 channels are expressed in both pyramidal neurons and interneurons (*Cooper et al., 2001*). In pyramidal neurons, KCNQ2/3 channels primarily control spike frequency adaptation, a quiescence period neurons enter following a brief train of activity (*Peters et al., 2005*; *Soh et al., 2014*). However, our knowledge of KCNQ2/3 function in interneurons is limited. This gap in knowledge is partly because KCNQ2/3 function is most easily observed in neurons that undergo pronounced spike frequency adaptation, a characteristic not traditionally found in interneurons (*Pelkey et al., 2017*).

It is currently unknown whether loss of KCNQ2/3 function in interneurons would have effects on overall network activity, and if it does, whether it would lead to a dampening or an increase in excitability. Previous work suggests that loss of *KCNQ2/3* function in interneurons would likely elevate their excitability (*Lawrence et al., 2006*; *Pelkey et al., 2017*), which in turn would increase GABAergic inhibition and decrease network excitability. However, at early times in development GABA is

depolarizing due to shifted chloride equilibrium potential (*Le Maguaresse and Monyer, 2013*). Previous work found that neonatal administration of bumetanide, which changes GABA receptor activity from depolarizating to hyperpolarizing, prevents seizures in mice with a global loss of KCNQ2 channels (*Marguet et al., 2015*). This may be evidence that loss of KCNQ2/3 function at this time increases network excitability through changes in interneuron GABA signaling, but the authors did not directly examine synaptic activity. Further, it is entirely unknown what effects KCNQ2/3 dysfunction in interneurons has on network excitability in juvenile and mature mice.

In this work, we developed mice lacking KCNQ2 and KCNQ3 specifically in interneurons to address their possible role at the cellular and network activity. We found that KCNQ2/3 channels regulate interneuron properties in a cell type-specific manner. Deletion of KCNQ2/3 channels primarily impacts the firing properties of PV$^+$ interneurons, but not SST$^+$ interneurons. We also find that interneuronal loss of KCNQ2/3 or KCNQ2 channels increases excitatory transmission between pyramidal neurons, perhaps as homeostatic compensation for the increased GABAergic signaling we observe.

## Results and discussion

To investigate the role of KCNQ2/3 channels in interneurons, we developed mouse lines that lack KCNQ2/3 channels specifically in parvalbumin-positive (PV$^+$) and somatostatin-positive (SST$^+$) interneurons, cell types known to express KCNQ2/3 channels (*Cooper et al., 2001*; *Lawrence et al., 2006*). We crossed *Kcnq2* or *Kcnq3* floxed mice (*Kcnq2$^{f/f}$* and *Kcnq3$^{f/f}$*) to *Nkx2-1$^{cre}$* mice (*Xu et al., 2008*). The *Nkx2-1$^{cre}$* driver is expressed starting early in development (~embryonic day 10.5) in SST$^+$ and PV$^+$ interneurons, allowing us to study the impact of KCNQ2/3 channel ablation in young and juvenile neurons. In order to identify the Cre-expressing PV$^+$ and SST$^+$ interneurons, we crossed these mice to a reporter line (Ai9) that expresses the fluorescent protein tdTomato in cells in which Cre recombinase has been active. We designated these mice as IN:Kcnq2/3 null (*Nkx2-1$^{cre}$;Kcnq2$^{f/f}$; Kcnq3$^{f/f}$;*Ai9).

We first examined whether ablation of KCNQ2/3 channels from interneurons led to changes in the excitatory and inhibitory drive of CA1 pyramidal neurons. Based on previous work (*Pelkey et al., 2017*), we predicted that loss of KCNQ2/3 channels from PV$^+$/SST$^+$ interneurons would lead to increased inhibitory but not excitatory transmission. We recorded spontaneous inhibitory postsynaptic currents (sIPSCs) and excitatory postsynaptic currents (sEPSCs) from CA1 pyramidal neurons of IN:Kcnq2/3 null mice. These measurements are commonly used to detect synaptic input. We first focused our analysis on the second week of life, as KCNQ2/3 dysfunction is primarily associated with neonatal epilepsy (*Oyrer et al., 2018*). Consistent with KCNQ2/3 channel expression in PV$^+$/SST$^+$ interneurons (*Pelkey et al., 2017*), the loss of these channels increased sIPSC frequency (*Figure 1a*; mean frequency: control 6.4 ± 0.5 Hz, n = 15; IN:Kcnq2/3 null 8.8 ± 1.1 Hz, n = 10; df = 23 t = −2.22 p=0.036 two-tailed Student's t-test). Importantly, ablation of *Kcnq2/3* from PV$^+$/SST$^+$ also increased sEPSC frequency in CA1 pyramidal neurons (*Figure 1a*; mean frequency: control 2.01 ± 0.16 Hz, n = 14; IN:Kcnq2/3 null 3.01 ± 0.35 Hz, n = 8; df = 20 t = −2.92 p=0.0085 Student's t-test). This frequency increase was accompanied by a three-fold change in the miniature EPSC (mEPSC) frequency (*Figure 1b*; mean frequency: control 0.71 ± 0.11 Hz, n = 9; IN:Kcnq2/3 null 2.2 ± 0.4 Hz, n = 6; p=0.002 df = 13 t = −3.86 Student's t-test), suggesting that elevating interneuron excitability led to secondary changes in excitatory synaptic drive. We did not find any changes in the miniature IPSC (mIPSC) frequency (*Figure 1b*; mean frequency: control 5.25 ± 0.48 Hz, n = 12; IN:Kcnq2/3 null 5.8 ± 0.6 Hz, n = 9; df = 19 t = −0.81 p=0.43 Student's t-test), suggesting that the effect on sIPSCs was due to elevated interneuron excitability. The effects on the mEPSC frequency were likely a result of synaptic homeostasis in an effort to maintain a constant excitation and inhibition ratio. This is because an increase in mEPSCs, and consequently sEPSCs, would counteract the increases in the sIPSC frequency. Indeed, the sEPSC/sIPSC ratio (mean ratio: control 0.34 ± 0.03, n = 14; IN:Kcnq2/3 null 0.36 ± 0.04, n = 8), unlike the mEPSC/mIPSC ratio (mean ratio control 0.17 ± 0.04, n = 9; IN:Kcnq2/3 null 0.38 ± 0.05, n = 5; df = 12 t = −3.51 p=0.004 unpaired Student's t-test), remained the same in neurons with or without PV$^+$/SST$^+$ KCNQ2/3 channels. Importantly, the effects on the mEPSCs were specific to changes in interneuron excitability, as we did not find any significant changes in the mEPSC frequency in CA1 pyramidal neurons from *Kcnq2* pyramidal neuron knockout mice (*Emx1$^{Cre}$;Kcnq2$^{f/f}$*, designated as PYR:Kcnq2 null) (*Figure 1*). Rather, in PYR:Kcnq2 null neurons

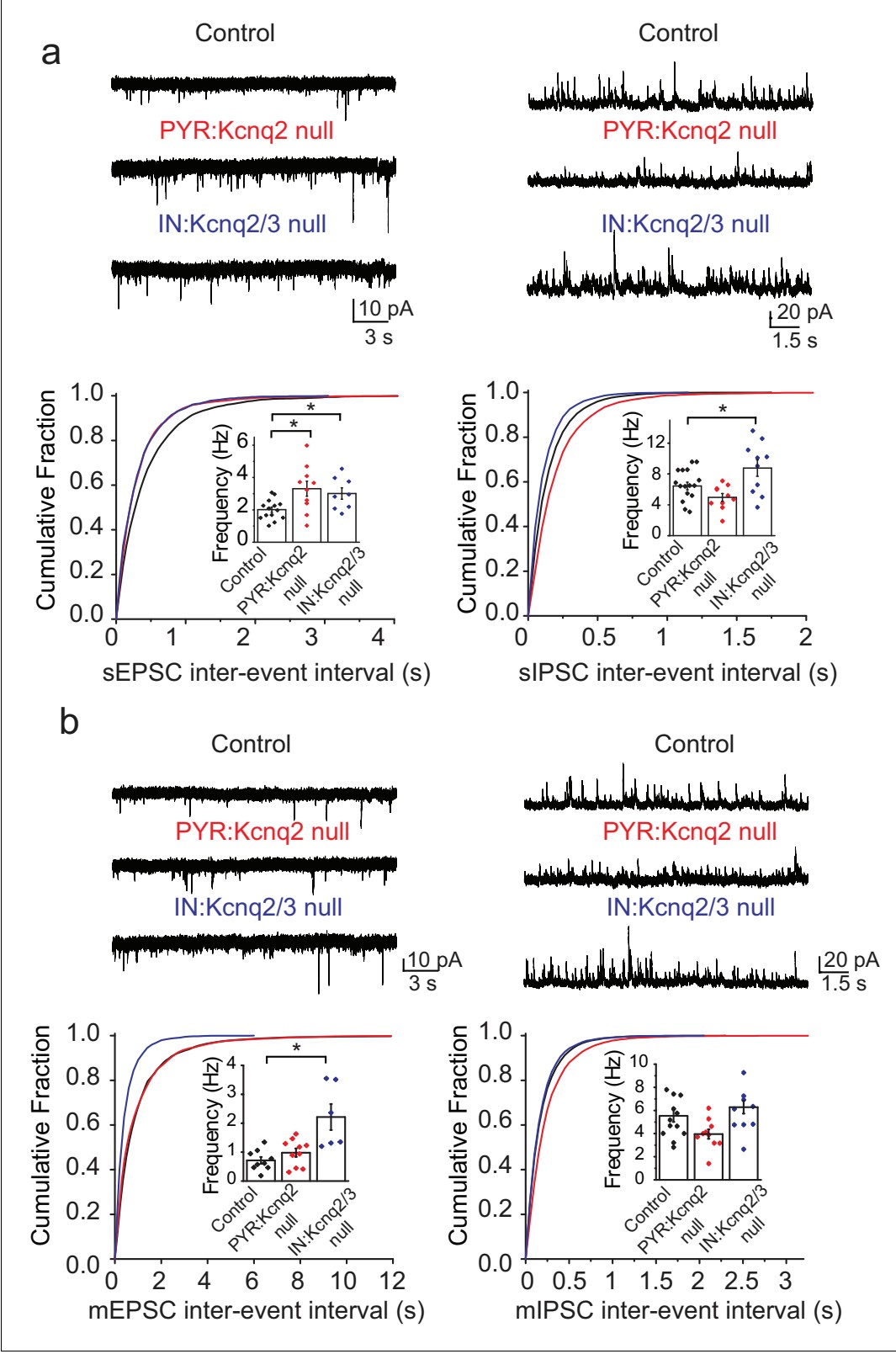

**Figure 1.** Ablation of *Kcnq2/3* channels from PV+/SST+interneurons leads to increased excitatory transmission. (a) Top, representative sEPSC and sIPSC traces recorded from mouse CA1 pyramidal neurons (P15–P19) in acute hippocampal slices from control (number of animals = 8) and either *Kcnq2/3* interneuron (IN:Kcnq2/3 null; number of animals = 4) or *Kcnq2* pyramidal neuron conditional knockout (PYR:Kcnq2 null; number of animals = 3) mice. Bottom, cumulative distribution plots of sIPSC and sEPSC inter-event intervals recorded in pyramidal neurons of IN:Kcnq2/3 null and PYR:Kcnq2 null

*Figure 1 continued on next page*

*Figure 1 continued*

mice. Insets: summary graphs of average inter-event frequency. Statistical comparisons were performed with a one-way analysis of variance (ANOVA; p<0.05) followed by a Fisher LSD post-hoc test (*: p<0.05, **: p<0.001). For comparing sEPSC frequency: ANOVA $F_{(2,29)}$ =5.168, p=0.012; for control vs. PYR:Kcnq2 null p=0.0051; for control vs. IN:Kcnq2/3 null p=0.038. (b) Top, representative mEPSC and mIPSC traces recorded from mouse CA1 pyramidal neurons from control (number of animals = 6), IN:Kcnq2/3 null (number of animals = 3) or PYR:Kcnq2 null (number of animals = 4) mice, respectively. Bottom, cumulative distribution plots of mIPSCs and mEPSCs inter-event intervals recorded in pyramidal neurons of IN:Kcnq2/3 null and PYR:Kcnq2 null mice. Insets: summary graphs of average inter-event frequency. Statistical comparisons were performed with one-way ANOVA followed by Fisher LSD post-hoc test (*: p<0.05, **: p<0.001). For comparing sEPSC frequency ANOVA $F_{(2,22)}$ =10.74, p=0.0006; for control vs. PYR:Kcnq2 null p=0.3764; for control vs. IN:Kcnq2/3 null p=0.0002. Each data point represents recording from one pyramidal neuron. Data in summary graphs are represented as mean and s.e.m.

DOI: https://doi.org/10.7554/eLife.38617.002

The following source data is available for figure 1:

**Source data 1.** Source data for *Figure 1*.

DOI: https://doi.org/10.7554/eLife.38617.003

only the spontaneous frequency was elevated (*Figure 1*; mean frequency: control 2.01 ± 0.16 Hz, n = 14; PYR:Kcnq2 null 3.3 ± 0.46 Hz, n = 10; df = 22 t = −2.99 p=0.0067 Student's t-test), consistent with the role of KCNQ2/3 channels in controlling pyramidal neurons' axonal excitability (*Battefeld et al., 2014*).

Are the synaptic changes due to loss of KCNQ2/3 from PV[+], SST[+], or both types of interneurons? To address this question, we tested whether ablation of KCNQ2/3 channels alters the excitability of PV[+] and SST[+] cells in mouse CA1. We distinguished these neurons based on their intrinsic excitability and firing properties (*Figure 2*). We found that ablation of *Kcnq2/3* from PV[+]-like interneurons led to an increase in the number of action potentials following suprathreshold depolarizing current pulses (*Figure 2a*). In contrast, ablation of *Kcnq2/3* from SST[+]-like interneurons did not change their firing properties (*Figure 2a*). We obtained similar results for L2/3 interneurons, suggesting that KCNQ2/3 function is critical for PV[+] interneuron properties across multiple forebrain regions (*Figure 2a*). To confirm our finding that KCNQ2/3 alters PV[+] interneuron excitability, we tested the effect of ablating *Kcnq2* and *Kcnq3* from PV[+] interneurons by developing *Pvalb[cre];Kcnq2[f/f]/Kcnq3[f/f]; Ai9* (designated as PV:Kcnq2/3 null) mice. Indeed, ablation of *Kcnq2* and *Kcnq3* increased the firing rate of PV[+] neurons in the CA1 region (*Figure 2b*). The increased excitability in PV[+] interneurons might have been due to their increased input resistance of *Kcnq2/3* null neurons (*Figure 2b*; mean $R_{IN}$: control 166 ± 16 MΩ, n = 15; PV:Kcnq2/3 null 257 ± 33 MΩ, n = 14; df = 27 t = −2.54 p=0.017 unpaired Student's t-test). However, as application of the pan-KCNQ blocker XE-991 did not change the input resistance in control PV[+] interneurons (*Figure 2—figure supplement 3*; $R_{IN}$: control 169 ± 20 MΩ,+XE-991 143 ± 20 MΩ, n = 6; t = 1.355 df = 5 p=0.233), it's unclear whether the change in input resistance in *Kcnq2/3* null neurons is due to direct loss of KCNQ2/3 activity or an indirect effect. Next, we repeated our experiments in *SST[cre];Kcnq2[f/f]/Kcnq3[f/f];Ai9* neurons (*Figure 2b*; refer to as SST:Kcnq2/3 null). Consistent with our earlier finding, ablation of *Kcnq2/3* from SST[+] interneurons did not change their firing properties or input resistance (mean $R_{IN}$: control 360.28 ± 34 MΩ, n = 6; SST:Kcnq2/3 null 327 ± 64 MΩ, n = 8; df = 12 t=−0.42 p=0.68 unpaired Student's t-test).

Our data suggest that ablation of KCNQ2/3 channel activity from PV[+] interneurons results in elevated interneuron excitability, which in turn could lead to remodeling of fast excitatory synaptic activity. As KCNQ2 pathogenic variants are much more frequent than KCNQ3 variants (*Brenner and Wilcox, 2012*), we also examined whether ablation of *Kcnq2* alone could lead to similar effects. Indeed, we found that loss of KCNQ2 channels from PV[+] interneurons increased their intrinsic excitability (*Figure 3a*) and the frequency of sEPSCs in pyramidal neurons (*Figure 3c*; *Pvalb-Cre;Kcnq2[+/+];Ai9* 1.165 ± 0.12 Hz, n = 13; *Pvalb-Cre;Kcnq2[f/f];Ai9*: 2.108 ± 0.22 Hz, n = 19; p=0.0098 Mann-Whitney test). The observed changes in the sEPSC frequency were not due to changes in the intrinsic excitability of pyramidal neurons (*Figure 3b*). To determine whether ablation of *Kcnq2* from PV[+]-expressing interneurons affected the excitatory drive *in vivo*, we administered picrotoxin, a GABA_A receptor antagonist. We implanted mice with bilateral frontal and parietal electrodes and validated generalized epileptiform activity coinciding with Racine scale Stage five seizures induced by intraperitoneal injection of picrotoxin (10 mg/kg) (*Figure 4a*). Thereafter, we video-recorded the latency to

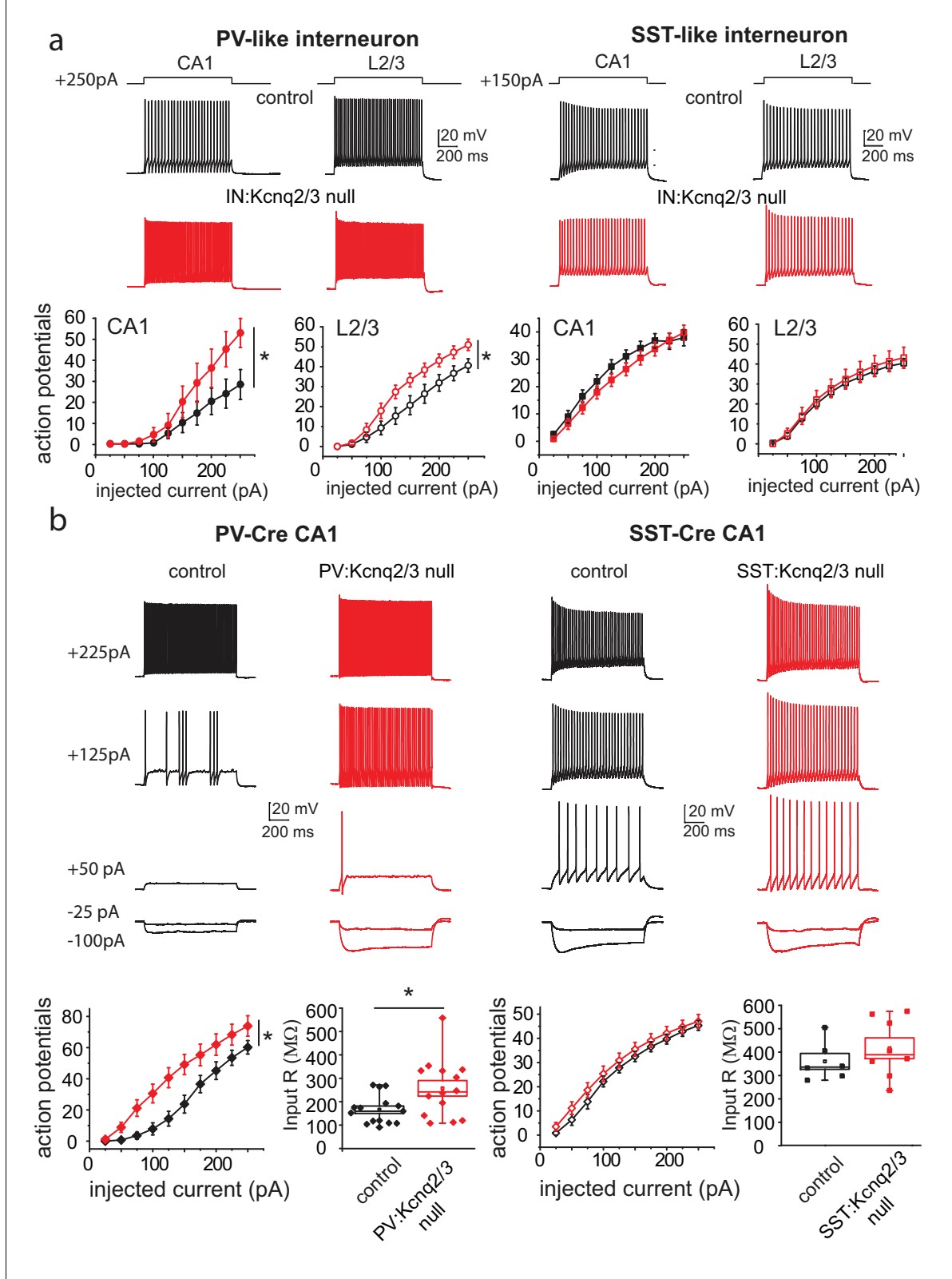

**Figure 2.** Loss of KCNQ2/3 activity leads to increased excitability of PV+interneurons. (a) Top, representative voltage responses from a +150 pA current injection step (1 s) in PV- and SST-like interneurons in either the CA1 region of the hippocampus (P12–P17) or L2/3 of the somatosensory cortex (P8–P11). For L2/3 recordings, cells were also confirmed by immunoreactivity against SST antibodies. Bottom, summary graphs showing the effect of deleting KCNQ2 and KCNQ3 channels on action potential number from CA1 PV-like (control n = 8/6; IN:Kcnq2/3 null n = 8/5), SST-like (control n = 19/

*Figure 2 continued on next page*

*Figure 2 continued*

8; IN:Kcnq2/3 null n = 8/4), and L2/3 (PV-like: control n = 10/7; IN:Kcnq2/3 null n = 8/4; SST-like: control n = 10/6; IN:Kcnq2/3 null n = 5/4) interneurons (Vh=-75 to −77 mV). For CA1 PV-like cells (P16–P25), $F_{(9,126)}$=2.849, p=0.0043; for L2/3 PV-like cells, $F_{(9,144)}$=3.845, p=0.0002; for CA1 SST-like cells (P15–P19), $F_{(9,225)}$=0.601, p=0.7955; and for L2/3 SST-like cells, $F_{(9,117)}$=0.326, p=0.965. Significance was determined using a two-factor mixed ANOVA. See *Figure 2—figure supplement 1* showing that indeed SST cells express KCNQ2 and KCNQ3 mRNA. (**b**) Top, representative voltage responses to a series of current injection steps (1 s) in PV$^+$ and SST$^+$ interneurons in the CA1 region of the hippocampus (Vh=-75 to −77 mV). Bottom left, summary graph showing the effect of deleting KCNQ2 and KCNQ3 channels on action potential number from CA1 PV$^+$ cells (control n = 15/8; PV:Kcnq2/3 null n = 14/7; $F_{(9,243)}$=3.558 with p=0.0004). Middle left, summary graph showing that loss of KCNQ2/3 channels decreases PV$^+$ input resistance (control, n = 15/8; PV:Kcnq2/3 null, n = 14/7; df = 27 t=−2.54 p=0.017 unpaired Student's t-test). See also *Figure 2—figure supplement 2* regarding PV$^+$ *Kcnq2/3* null neurons diversity of intrinsic properties. Middle right, summary graph showing the effect of deleting KCNQ2 and KCNQ3 channels on action potential number from CA1 SST$^+$ cells (control n = 6/2; SST:Kcnq2/3 null n = 8/4; $F_{(9,108)}$=0.729 with p=0.6814). Bottom right, summary graph showing loss of KCNQ2/3 channels did not decrease SST$^+$ input resistance (control n = 6/2; SST:Kcnq2/3 null, n = 8/4; df = 12 t=−0.42 p=0.68 unpaired Student's t-test). 'n' designates number of cells followed by number of animals. Each data point represents recording from one neuron. Data in summary graphs are represented as mean and s.e.m.

DOI: https://doi.org/10.7554/eLife.38617.004

The following source data and figure supplements are available for figure 2:

**Source data 1.** Source data for *Figure 2*.

DOI: https://doi.org/10.7554/eLife.38617.010

**Figure supplement 1.** FISH shows presence of KCNQ2 and KCNQ3 in SST$^+$interneurons.

DOI: https://doi.org/10.7554/eLife.38617.005

**Figure supplement 2.** *PV:Kcnq2/3* null interneurons could differ in their intrinsic excitability properties.

DOI: https://doi.org/10.7554/eLife.38617.006

**Figure supplement 2—source data 1.** Source data for *Figure 2—figure supplement 2*.

DOI: https://doi.org/10.7554/eLife.38617.007

**Figure supplement 3.** The pan-KCNQ blocker XE991 increases PV$^+$interneuron excitability.

DOI: https://doi.org/10.7554/eLife.38617.008

**Figure supplement 3—source data 1.** Source data for *Figure 2—figure supplement 3*.

DOI: https://doi.org/10.7554/eLife.38617.009

seizure onset without implantation. All of the mice lacking *Kcnq2*, including Pvalb;*Kcnq2* heterozygous mice, reached Stage five seizure activity within 30 min, with a significantly reduced latency to seizure onset when compared to Pvalb;*Kcnq2$^{+/+}$* mice (*Figure 4b*). These data confirm that loss of KCNQ2 channel activity from interneurons can lead to excitatory network hyperexcitability *in vivo*.

In summary, our *ex vivo* and *in vivo* work reveals that changes in interneuron excitability induced by loss of KCNQ2/3 channel activity in PV$^+$ interneurons could change the course of excitatory network development. Multiple studies have shown that early in development GABA-mediated transmission partly drives excitatory synapse maturation and formation in pyramidal neurons. This is because early in development pyramidal neurons have a depolarized GABA equilibrium potential (*Le Magueresse and Monyer, 2013*; *Wang and Kriegstein, 2011*), allowing GABA to provide a much-needed depolarizing signal to relieve magnesium block from NMDA receptors and in turn promote AMPA receptor unsilencing and synaptic input maturation. Indeed, recent work showed that pharmacologically blocking potassium channels with 4-AP to drive GABA release on newborn neurons led to robust formation of excitatory synaptic inputs, allowing for the integration of newborn cells to excitatory synaptic networks (*Chancey et al., 2013*). We thus speculate a similar mechanism might in play in our mice. Future work is needed to directly test this hypothesis.

Importantly, our work suggests a unique mechanism compared to earlier work showing that loss of potassium channel activity critical for interneuron firing behavior leads only to depolarization block (*Lau et al., 2000*) and hyperexcitability through disinhibition. Therefore, our work raises the possibility that potassium channel dysfunction that elevates GABAergic transmission could also lead to remodeling of excitatory transmission in the cortex. Such changes might contribute to the severe neurodevelopmental effects seen in patients with potassium channel epileptic encephalopathy.

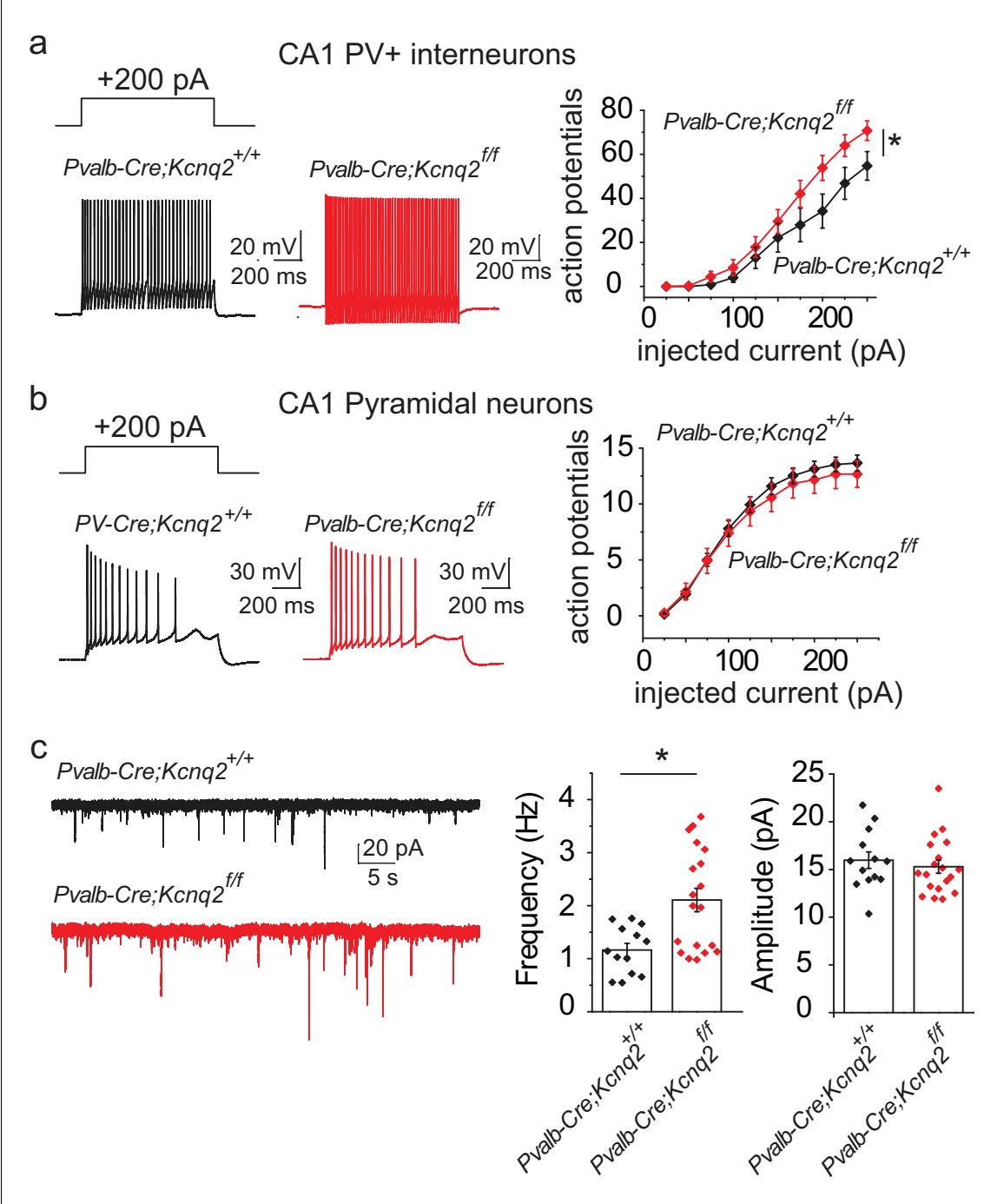

**Figure 3.** Ablation of *Kcnq2* from PV[+] interneurons leads to increased excitatory transmission in pyramidal neurons. For simplicity we refer *Pvalb-Cre; Kcnq2*;Ai9 mice in the figure as *Pvalb-Cre;Kcnq2^{f/f}* or *Pvalb-Cre;Kcnq2^{+/+}*. (**a**) Left, representative voltage responses from a + 200 pA current injection step (1 s; Vh= −75 to −77 mV) in PV[+] interneurons from the CA1 region of the hippocampus (P23–P25). Right, summary graph showing the effect of deleting *Kcnq2* on action potential number from PV[+] interneurons (*Pvalb-Cre;Kcnq2^{+/+}*;Ai9 n = 15/4; *Pvalb-Cre;Kcnq2^{f/f}*;Ai9 n = 14/3; $F_{(9,243)}$=3.558, p=0.0004). Significance was determined using a two-factor mixed ANOVA. (**b**) Left, representative voltage responses from a + 200 pA current injection step (1 s; Vh=-75mV) in pyramidal neurons from the CA1 region of the hippocampus (P30–P32). Right, summary graph showing the effect of deleting *Kcnq2* from pyramidal neurons in action potential number (*Pvalb-Cre;Kcnq2^{+/+}*;Ai9 n = 15/3; *Pvalb-Cre;Kcnq2^{f/f}*;Ai9 n = 12/2; $F_{(9,225)}$=0.4891, p=0.88). Significance was determined using a two-factor mixed ANOVA. (**c**) Left, representative sEPSC traces recorded from CA1 pyramidal neurons (P32–P35) in acute hippocampal slices from control and *Kcnq2* null PV[+] interneurons. Right, summary bar graphs of sEPSC frequency (*Pvalb-Cre;Kcnq2^{+/+}*;Ai9 1.165 ± 0.12 Hz, n = 13/4; *Pvalb-Cre;Kcnq2^{f/f}*;Ai9: 2.108 ± 0.22 Hz, n = 19/3; p=0.0098 Mann-Whitney test) and amplitude (*Pvalb-Cre;Kcnq2^{+/+}*;Ai9 15.9 ± 0.86 pA, n = 13/4; *Pvalb-Cre;Kcnq2^{f/f}*;Ai9: 15.3 ± 0.68 pA, n = 19/3; DF = 30 t = 0.62 p=0.5394). Statistical comparisons were performed with two-

*Figure 3 continued on next page*

*Figure 3 continued*

tailed unpaired Student's t-test or Mann-Whitney when the variance between the two groups was significantly different. 'n' designates number of cells followed by number of animals. Each data point represents recording from one neuron. Data in summary graphs are represented as mean and s.e.m.

DOI: https://doi.org/10.7554/eLife.38617.011

The following source data is available for figure 3:

**Source data 1.** Source data for *Figure 3*.

DOI: https://doi.org/10.7554/eLife.38617.012

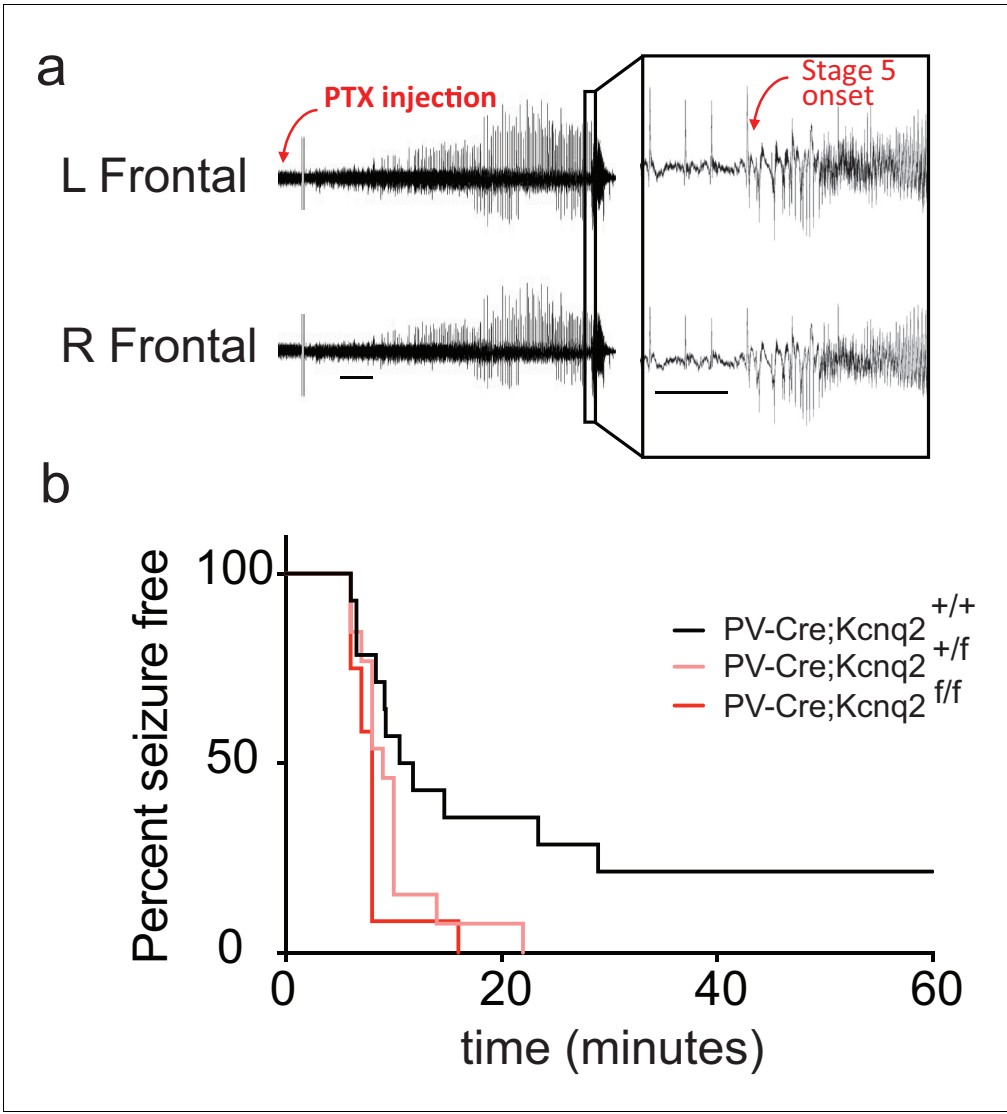

**Figure 4.** *In vivo* hyperexcitability with loss of *Kcnq2* in PV-expressing interneurons. (**a**) Using simultaneous video-EEG monitoring, Stage five onset was defined as the latency to rearing and falling with forelimb clonus, associated with bilateral epileptiform activity on EEG. (**b**) Loss of one or both *Kcnq2* alleles in PV-expressing interneurons led to significantly reduced latency to seizure-onset (p=0.0370 and p=0.0047, respectively, Log-rank (Mantel-Cox) test; *Pvalb-cre;Kcnq2*<sup>+/+</sup> n = 14; *Pvalb-cre;Kcnq2*<sup>f/+</sup> n = 13; *Pvalb-cre;Kcnq2*<sup>f/f</sup> n = 12). 'n' designates number of animals. Bar = 1 min, inset, 5 s.

DOI: https://doi.org/10.7554/eLife.38617.013

The following source data is available for figure 4:

**Source data 1.** Source data for *Figure 4*.

DOI: https://doi.org/10.7554/eLife.38617.014

# Materials and methods

## Key resources table

| Reagent type (species) or resource | Designation | Source or reference | Identifiers | Additional information |
|---|---|---|---|---|
| Gene (Mus musculus) | Kcnq2 | NA | NCBI_Gene:16536; MGI:1309503 | |
| Gene (Mus musculus) | Kcnq3 | NA | NCBI_Gene:110862; MGI:1336181 | |
| Strain, strain background (M. musculus, $Emx1^{IRES}$Cre, C57BL/6J background) | B6.129S2-Emx1$^{tm1(cre)Krj}$/J | PMID: 12151506 | RRID:IMSR_JAX:005628 | |
| Strain, strain background (M. musculus, Nkx2.1-Cre, C57BL/6J background) | C57BL/6J-Tg (Nkx2-1-cre)2Sand/J | PMID: 17990269 | RRID:IMSR_JAX:008661 | |
| Strain, strain background (M. musculus, Sst-IRES-Cre, C57BL/6J background) | Sst$^{tm2.1(cre)Zjh}$/J | PMID: 21943598 | RRID:IMSR_JAX:013044 | |
| Strain, strain background (M. musculus, Pvalb-Cre, C57BL/6J background) | B6;129P2-Pvalb$^{tm1(cre)Arbr}$/J | PMID: 15836427 | RRID:IMSR_JAX:008069 | |
| Strain, strain background (M. musculus, tdTomato reporter Ai9, C57BL/6J background) | B6.Cg-Gt(ROSA) 26Sor$^{tm9(CAG-tdTomato)Hze}$/J | PMID: 20023653 | RRID:IMSR_JAX:007909 | |
| Strain, strain background (M. musculus, $Kcnq2^{f/f}$, C57BL/6J background) | Kcnq2$^{f/f}$ | PMID: 24719109 | N-A | |
| Strain, strain background (M. musculus, $Kcnq3^{f/f}$, C57BL/6J background) | Kcnq3$^{f/f}$ | PMID: 24719109 | N-A | |
| Antibody | Alexa fluor 488 streptavidin | Invitrogen | Invitrogen:S32354; RRID:AB_2315383 | (1:500) |
| Antibody | anti-Lucifer yellow (rabbit polyclonal) | Invitrogen | Invitrogen:A5750; RRID:AB_2536190 | (1:500) |
| Antibody | anti-Somatostatin (rat monoclonal) | Millipore | Millipore:MAB354; RRID:AB_2255365 | (1:250) |
| Antibody | Alexa fluor 488 anti-rabbit secondary (goat polyclonal) | Invitrogen | Invitrogen:A11034; RRID:AB_2576217 | (1:500) |
| Antibody | Alexa fluor 647 anti-rat secondary (goat polyclonal) | Invitrogen | Invitrogen:A21247; RRID:AB_141778 | (1:500) |
| Sequence-based reagent | somatostatin mRNA probe (mouse); Mm-Sst-C1 | ACDBio | Cat#:404631 | (1:50) |
| Sequence-based reagent | parvalbumin mRNA probe (mouse); Mm-Pvalb-C1 | ACDBio | Cat#:421931 | (1:50) |
| Sequence-based reagent | tdTomato mRNA probe (mouse); Mm-tdTomato-C3 | ACDBio | Cat#:317041-C3 | (1:50) |
| Sequence-based reagent | Kcnq2 mRNA probe (mouse); Mm-Kcnq2-O1 | ACDBio; this paper | Cat#:300031-C2 | (1:50); custom made probe that targets exons 2–5 of Kcnq2 |

*Continued on next page*

*Continued*

| Reagent type (species) or resource | Designation | Source or reference | Identifiers | Additional information |
|---|---|---|---|---|
| Sequence-based reagent | *Kcnq3* mRNA probe (mouse); Mm-*Kcnq3*-O1 | ACDBio; this paper | Cat#:300031-C3 | (1:50); custom made probe that targets exons 2–5 of *Kcnq2* |
| Commercial assay or kit | RNAscope Fresh Frozen Multiplex Fluorescent kit | ACDBio | Cat#:320851 | |
| Chemical compound, drug | CNQX | Abcam | ab120017 | |
| Chemical compound, drug | D-AP5 | Abcam | ab120003 | |
| Chemical compound, drug | Picrotoxin | Abcam | ab120315 | |
| Chemical compound, drug | Tetrodotoxin; TTX | Abcam | ab120054 | |
| Chemical compound, drug | XE-991 | Abcam | ab120089 | |
| Chemical compound, drug | Lucifer Yellow | Sigma | Cat#:B4261 | (0.1%) |
| Chemical compound, drug | Biocytin | Molecular Probes | Cat#:L1177 | (0.05%) |
| Software, algorithm | Prism 7 | GraphPad | RRID:SCR_002798 | Version 7.03 |
| Software, algorithm | Clampfit 10 | Molecular Devices | RRID:SCR_011323 | |
| Software, algorithm | Minianalysis | Synaptosoft | RRID:SCR_002184 | |
| Software, algorithm | Origin 8 Pro | OriginLab | RRID:SCR_014212 | Version 8.0951 |
| Software, algorithm | ImageJ | NIH | RRID:SCR_003070 | Version 2.0.0 |

## Ethics statement

All experiments were performed according to the guidelines described in the National Institutes of Health Guide for the Care and Use of Laboratory Animals and were approved by the Institutional Animal Care and Use Committee of the University of Connecticut, Storrs (Protocol: A16-031) and of Baylor College of Medicine (Protocol: AN-6600).

## Animals

For our experiments, we used both male and female mice. To generate interneuron or pyramidal neuron specific *Kcnq2* and *Kcnq3* conditional knockout mice, we crossed our previously generated *Kcnq2*$^{f/f}$, *Kcnq3*$^{f/f}$, or *Kcnq2*$^{f/f}$/*Kcnq3*$^{f/f}$ mice with the following Cre driver lines obtained from Jackson laboratories. For pyramidal neuron *Kcnq2* ablation we used B6.129S2-*Emx1*$^{tm1(cre)Krj}$/J (Jax stock # 005628), whereas for interneuron *Kcnq2* and *Kcnq3* cell type-specific knockout mice we used C57BL/6J-Tg(Nkx2-1-cre)2Sand/J (Jax stock #008661), *Sst*$^{tm2.1(cre)Zjh}$/J (Jax stock #013044), or B6;129P2-*Pvalb*$^{tm1(cre)Arbr}$/J (Jax stock #008069). To visualize cells that underwent recombination we also crossed the interneuron specific mouse lines with tdTomato (Ai9) reporter mice B6.Cg-*Gt (ROSA)26Sor*$^{tm9(CAG-tdTomato)Hze}$/J (Jax stock #007909). The *Pvalb* and *Kcnq3* are on chromosome 15 too close to each other (~7 centi-morgan separation) for rare recombination events to occur making it difficult to generate *Pvalb-Cre;Kcnq3*$^{f/f}$ mice. However, we were successful in generating a *Pvalb-Cre;Kcnq3*$^{f/f}$ mouse which we then crossed to *Kcnq2*$^{f/f}$ mice to generate a *Pvalb-Cre;Kcnq3*$^{f/f}$; *Kcnq2*$^{f/+}$. Constitutive *Kcnq3*$^{-/-}$ mice were described previously (*Kim et al., 2016*). Mice were genotyped by PCR using the protocol described on the Jax.org website. Data collection and analysis of the *in vivo* seizure experiments were performed blind.

## Slice electrophysiology

Mice (P8-P35) were anesthetized with isoflurane and rapidly decapitated. The brain was quickly removed and placed in ice-cold sucrose based cutting solution consisting of the following: 25 mM NaHCO$_3$, 200 mM sucrose, 10 mM glucose, 2.5 mM KCl, 1.3 mM NaH$_2$PO$_4$, 0.5 mM CaCl$_2$, and 7

mM MgCl$_2$. Coronal or Transverse hippocampus slices including surrounding structures were cut at 300 μm using a vibrating microtome (Microm HM 650V-Thermo Fisher Scientific; or Leica VT1200S). Slices were then transferred in artificial cerebrospinal fluid (ACSF) consisting of the following (in mM): 125 NaCl, 26 NaHCO$_3$, 2.5 KCl, 1 NaH$_2$PO$_4$, 1.3 MgCl$_2$, 2.5 CaCl$_2$, and 12 glucose and equilibrated at 35°C for 30 min, and then maintained at room temperature for at least 1 hr before electrophysiological recordings. Cutting solution and ACSF were saturated with 95% O$_2$ and 5% CO$_2$. All experiments were performed at room temperature. Whole-cell recordings were obtained using borosilicate glass electrodes having resistances of 2 to 4 MΩ. For current clamp whole cell recordings we used an internal consisted of the following (in mM): 130 potassium methylsulfate (or potassium gluconate), 10 KCl, 5 Tris-phosphocreatine, 10 HEPES, 4 NaCl, 4 Mg$_2$ATP, and 0.4 Na$_4$GTP. The pH was adjusted to 7.2 to 7.3 with KOH. CNQX (4 μM), D-AP5 (10 μM), and picrotoxin (100 μM) (Abcam) were added in all current clamp experiments (*Figure 2* and *Figure 3a*) to block AMPA-mediated, NMDA-mediated, and GABA-mediated synaptic transmission, respectively. For synaptic activity recordings in voltage-clamp configuration we used an internal solution consisting of the following (in mM): 135 Cs-MeSO$_3$, 8 NaCl, 10 HEPES, 0.3 EGTA, 5 QX-314, 0.4 Na$_4$GTP, and 4 Mg$_2$ATP. The pH was adjusted to 7.2 with CsOH. To measure mEPSC and mIPSC, 1 μM TTX (Abcam) was present after recording spontaneous synaptic activity. We recorded s/mEPSCs in voltage-clamp mode at −70 mV, which is the reversal potential for GABA$_A$-mediated chloride currents when using a low internal chloride concentration. For s/mIPSCs were recorded at 0 mV the reversal of AMPA-mediated currents. For mEPSCs, we performed all recordings in the presence of 1 μM TTX to block action potentials. Recordings were performed using a Multiclamp 700B amplifier (Molecular Devices), low pass-filtered at 2 kHz, sampled at 10 kHz, and analyzed offline using either Prism 7 (Graphpad), Clampfit 10 (PCLAMP; Molecular Devices), Mini analysis program (Synaptosoft), or Origin eight pro (OriginLab).

## *In vivo* recordings

We implanted mice between the ages P30-45. For all mice we gave 48 hr post-operative recovery time before chemoconvulsive testing. Mice were implanted as previously described (*Maheshwari et al., 2017*). Briefly, mice were anesthetized with isoflurane (2–4% in O$_2$) anesthesia and surgically implanted with silver wire electrodes (0.005 inch diameter) inserted into the epidural space over the somatosensory cortex (1 mm posterior and 3 mm lateral to bregma) and frontal cortex (1 mm anterior and 1 mm lateral to bregma) bilaterally through cranial burr holes and attached to a microminiature connector cemented to the skull. The reference electrode was placed over the right cerebellum (1 mm posterior and 1 mm lateral to lambda) and the ground electrode was placed over the left cerebellum (1 mm posterior and 1 mm lateral to lambda).

## Post-hoc immuno staining

For L2/3 recordings in *Nxk2.1$^{IREScre}$Kcnq2$^{f/f}$;Kcnq3$^{f/f}$* mice, we verified SST positive cells by including in the recording solution either biocytin (0.05%; Sigma, cat# B4261) or lucifer yellow (0.1%, molecular probes, cat# L1177). Slices were subsequently fixed overnight in 4% paraformaldehyde in phosphate buffer (PB; pH 7.4) at 4°C and rinsed with PBS. Using standard immunofluorescence approach slices were incubated with 0.1% Triton X-100 and 10% normal goat serum to block non-specific binding. Slices were then incubated with Alexa Fluor 488 streptavidin (Invitrogen; cat# S32354) or rabbit anti-Lucifer yellow (Invitrogen; cat# A5750) along with rat anti-somatostatin (Millipore; cat# MAB354). We used Alexa Fluor 488 anti-rabbit (Invitrogen; cat# A11034) and Alexa 647 anti-rat (Invitrogen; cat# A21247) as secondary antibodies.

## Fluorescence In Situ Hybridization (FISH)

To prepare brain sections for FISH, mice were anesthetized with isoflurane and rapidly decapitated. Mouse brains (P19-22) were flash frozen embedded in OCT compound and coronal slices cryosectioned at a thickness of 14 μm and mounted on SuperFrost Plus Gold slides (Fisher Scientific). Sections were fixed in 4% paraformaldehyde for 15 min at 4°C, dehydrated in 50, 70, and 100% ethanol, and air-dried at room temperature. Fluorescent RNAscope in situ hybridization (ISH) was performed using an RNAscope Fresh Frozen Multiplex Fluorescent kit according to the manufacturer's protocol to perform target probe hybridization and signal amplification (Advanced Cell Diagnostics). Probes

were purchased from Advanced Cell Diagnostics: somatostatin mRNA, Mm-Sst-C1 (catalog #404631), parvalbumin mRNA, Mm-Pvalb-C1 (catalog #421931), tdTomato mRNA, Mm-tdTomato-C3 (catalog # 317041-C3), *Kcnq2* mRNA, *Mm-Kcnq2-O1* (catalog #300031-C2) and *Kcnq3* mRNA, Mm-Kcnq3-O1 (catalog #300031-C3). Probes for *Kcnq2* and *Kcnq3* were custom made targeting exons 2–5 and exons 2–4, respectively. Confocal images of FISH experiments were obtained using a Leica TSC Sp8 and confocal image files (lif) containing image stacks were loaded into ImageJ (version 2.0.0, NIH, RRID: SCR_003070).

## Analysis and quantification

Analytical tests were performed in Origin Pro (v8.0951) or GraphPad Prism (7.03) to calculate t tests, Mann-Whitney U, Log-rank (Mantel-Cox), or ANOVA with post hoc tests. All t-tests were two-tailed. Outliers were determined by Grubb's test (http://graphpad.com/ quickcalcs/Grubbs1.cfm). No statistical methods were used to predetermine sample sizes, but our sample sizes are similar to those reported in our previous publications.

## Acknowledgements

We thank members of our labs for comments on our manuscript. This work was supported by grants to AVT and AM.

## Additional information

### Funding

| Funder | Grant reference number | Author |
| --- | --- | --- |
| National Institutes of Health | NS073981 | Anastasios V Tzingounis |
| National Institutes of Health | NS101596 | Anastasios V Tzingounis |
| National Institutes of Health | NS096029 | Atul Maheshwari |

The funders had no role in study design, data collection and interpretation, or the decision to submit the work for publication.

### Author contributions

Heun Soh, Conceptualization, Data curation, Formal analysis, Supervision, Investigation, Methodology, Writing—original draft; Suhyeorn Park, Formal analysis, Investigation; Kali Ryan, Investigation; Kristen Springer, Data curation, Investigation; Atul Maheshwari, Conceptualization, Data curation, Formal analysis, Funding acquisition, Investigation, Writing—original draft, Writing—review and editing; Anastasios V Tzingounis, Conceptualization, Formal analysis, Supervision, Funding acquisition, Methodology, Writing—original draft, Project administration, Writing—review and editing

### Author ORCIDs

Atul Maheshwari (iD) https://orcid.org/0000-0003-3045-7901
Anastasios V Tzingounis (iD) https://orcid.org/0000-0002-4605-3437

### Ethics

Animal experimentation: All experiments were performed according to the guidelines described in the National Institutes of Health Guide for the Care and Use of Laboratory Animals and were approved by the Institutional Animal Care and Use Committee of the University of Connecticut, Storrs (Protocol#: A16-031) and of Baylor College of Medicine (Protocol#: AN-6600).

### Decision letter and Author response

Decision letter https://doi.org/10.7554/eLife.38617.017
Author response https://doi.org/10.7554/eLife.38617.018

## Additional files

### Supplementary files

• Transparent reporting form
DOI: https://doi.org/10.7554/eLife.38617.015

All data analysed during this study are included in the manuscript and supporting files. Data source files are also provided.

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
