## [Decision Letter]

Thank you for submitting your article "Deletion of KCNQ2/3 potassium channels from PV interneurons leads to homeostatic potentiation of excitatory transmission" for consideration by *eLife*. Your article has been reviewed by Gary Westbrook as the Senior Editor, a Reviewing Editor and four reviewers. The following individual involved in review of your submission has agreed to reveal his identity: Hillel Adesnik (Reviewer #2). The reviewers have discussed the reviews with one another and the Reviewing Editor has drafted this decision to help you prepare a revised submission.

Summary:

Soh et al., demonstrate that deletion of KCNQ2/3 specifically from GABAergic neurons – and specifically from PV interneurons – increases their excitability, increases inhibition onto pyramidal cells, but more surprisingly, increases excitation onto the pyramidal cell network. The increased excitation can only be interpreted as a non-cell autonomous effect that the authors plausibly attribute to 'synaptic homeostasis' – although this is not demonstrated directly. Formally demonstrating homeostasis would require showing that the firing rates of pyramidal cells in the mutant mice are comparable to that measured in wild types despite the overall increase in synaptic input in the mutant mice, something that could be tested in brain slices with electrical stimulation to the Schaffer collaterals or application of a solution that drives spontaneous activity. But the conclusion is reasonable even without this piece of evidence.

Perhaps most interesting is that deleting KCNQ2 channels from interneurons increases susceptibility to a chemoconvulsant, opposite of what might be simplistically expected from enhancement of interneuron excitability (and fitting with neonatal epilepsies associated with KCNQ2 mutations).

The slice electrophysiology demonstrating differences in excitability between control and knockout interneurons is thoroughly and rigorously performed. The experiments concisely and succinctly demonstrate relevant changes in electrical properties (current-clamp) upon removal of KCNQ2/3 channels, however the KCNQ2/3 currents are not directly measured in any way. Nonetheless, this is an interesting and important work, and with the revisions as indicated below, could make a substantial contribution.

The importance of this study is that it raises an intriguing notion that genetic mutations in KCNQ genes that give rise to epilepsy could be doing so not because of direct alterations to principal cell intrinsic excitability, but due to network changes associated with increased interneuron excitability.

Essential revisions:

There is a notable mis-match in the nature of the animals used for the in vitro vs. in vivo experiments. The in vitro experiments were done with mice lacking both KCNQ2 and KCNQ3 channels, while the in vivo experiments used mice lacking only KCNQ2 channels. The authors say that most human mutations involve KCNQ2. However, while these mice may be most relevant for comparing to human epilepsies, they are different from those used to explore the excitability differences in the in vitro experiments, leaving open the question of whether loss of KCNQ2 alone would have similar effects as loss of both. Thus, there is an issue linking the two parts of the manuscript (and one that the authors never address, partly by virtue of not having a separate Discussion section following the Results section). This issue could be fairly easily addressed if the authors tested the key in vitro results – enhanced excitability of PV-like interneurons and enhanced excitatory input to CA1 – in mice lacking only KCNQ2 in interneurons. One approach would be to repeat the core findings on increasing excitation and excitability in the PV-Cre;KCNQfl/fl mice.

[Editors' note: further revisions were requested prior to acceptance, as described below.]

Thank you for resubmitting your work entitled "Deletion of KCNQ2/3 potassium channels from PV interneurons leads to homeostatic potentiation of excitatory transmission" for further consideration at *eLife*. Your revised article has been favorably evaluated by Gary Westbrook (Senior Editor), a Reviewing Editor, and two reviewers.

The manuscript has been improved but there is one remaining issue that needs to be addressed before acceptance. Specifically, the shorthand nomenclature for the mutant animals could be quite confusing for readers. The authors say (Results and Discussion section) they will use a designation of "IN:KCNQ2/3 null" for the mice, but then proceed in both text and figures to refer to the animals as simply "IN:KCNQ2/3", which is quite confusing. Someone first looking at the paper by scanning the figures could easily assume that the authors are over-expressing KCNQ2/3 in interneurons or selectively restoring KCNQ2/3 in interneurons in an otherwise null mouse – very confusing. I would recommend using "IN:KCNQ2/3 null" everywhere.

---

## [Author Response]

Essential revisions:

There is a notable mis-match in the nature of the animals used for the in vitro vs. in vivo experiments. The in vitro experiments were done with mice lacking both KCNQ2 and KCNQ3 channels, while the in vivo experiments used mice lacking only KCNQ2 channels. The authors say that most human mutations involve KCNQ2. However, while these mice may be most relevant for comparing to human epilepsies, they are different from those used to explore the excitability differences in the in vitro experiments, leaving open the question of whether loss of KCNQ2 alone would have similar effects as loss of both. Thus, there is an issue linking the two parts of the manuscript (and one that the authors never address, partly by virtue of not having a separate Discussion section following the Results section). This issue could be fairly easily addressed if the authors tested the key in vitro results – enhanced excitability of PV-like interneurons and enhanced excitatory input to CA1 – in mice lacking only KCNQ2 in interneurons. One approach would be to repeat the core findings on increasing excitation and excitability in the PV-Cre;KCNQfl/fl mice.

We agree with the reviewers and have now performed the suggested experiments in *PV^+^;Kcnq2^f/f^;Ai9* and *PV^+^;Kcnq2^+/+^;Ai9* mice. We have included the additional results in the manuscript and also display them as a new Figure (Figure 3).

We now report that loss of *Kcnq2* from PV^+^ interneurons increased their intrinsic excitability (see Figure 3). We also recorded spontaneous EPSCs (sEPSC) in pyramidal neurons from the CA1 region from animals lacking *Kcnq2* in PV^+^ interneurons. In agreement with our previous data, the loss of *Kcnq2* led to an increase in the sEPSC frequency (control: 1.165 ± 0.12 Hz, n=13; *PV-Kcnq2* KO: 2.108 ± 0.221 Hz, n=19; P=0.0026). To ensure that the observed changes in the sEPSC frequency were not due to changes in pyramidal neurons intrinsic excitability we also recorded the firing properties of pyramidal neurons from the *PV^+^;Kcnq2^f/f^;Ai9* mice. We did not find any differences between the two groups (see new Figure 3). Thus, loss of either *Kcnq2* or *Kcnq2/3* from PV^+^ interneurons leads to elevated excitatory synaptic transmission in pyramidal neurons.

[Editors' note: further revisions were requested prior to acceptance, as described below.]

The manuscript has been improved but there is one remaining issue that needs to be addressed before acceptance. Specifically, the shorthand nomenclature for the mutant animals could be quite confusing for readers. The authors say (Results and Discussion section) they will use a designation of "IN:KCNQ2/3 null" for the mice, but then proceed in both text and figures to refer to the animals as simply "IN:KCNQ2/3", which is quite confusing. Someone first looking at the paper by scanning the figures could easily assume that the authors are over-expressing KCNQ2/3 in interneurons or selectively restoring KCNQ2/3 in interneurons in an otherwise null mouse – very confusing. I would recommend using "IN:KCNQ2/3 null" everywhere.

We agree that our nomenclature could be confusing. We have now used find and replace in the word document and to replace all IN:Kcnq2/3 with IN:Kcnq2/3 null. We also corrected the figures.

For consistency we also replaced PV:Kcnq2/3 with PV:Kcnq2/3 null, SST:Kcnq2/3 with SST:Kcnq2/3 null, and PV:Kcnq2 with PV:Kcnq2 null.

Finally, we found a typo in the manuscript which have now corrected. Kcnq3^-/-^ should have been labeled Kcnq3^f/f^. We have made this change now.